# TOWARDS PREDICATE-POWERED LEARNING

## ABSTRACT

The traditional approach to data-driven learning has become increasingly demanding in terms of its training data and computational resources. This work further develops a new paradigm of learning using predicates to reduce the need of data in learning. Among many recent efforts towards the same direction, *Learning using Statistical Invariants* (LUSI) has been proposed to be the new paradigm of learning. Building on top of LUSI and to break the "brute force" learning trend, we build towards a generalized theory of predicates and the invariants. The primary objective of this work is to propose an Extended Structure Risk Minimization (ESRM) paradigm with predicates, and provide a theoretical justification of the need for predicates in learning problems from both data complexity and model complexity perspectives. In this work, we show that predicates not only can aid in reducing the need for data in training, but they are also imperative for a highly efficient model.

Our primary contributions consist of the following: I) Proposing an extension to the structure risk minimization paradigm of learning, and II) Proving the efficacy of predicates in reducing both the data complexity and the model complexity.

## 1 INTRODUCTION

Classical learning theory is based on empirical risk minimization. This theoretical principle has been driving the development of machine learning research in the past few decades. While it has shown great success, it can also be considered as a brute force method Vapnik & Izmailov (2019). This data-driven approach can become exceedingly data and resource demanding, for example, large language models would cost millions of dollars to train. Recently, a new paradigm, *learning using statistical invariants*, has been proposed Vapnik & Izmailov (2019); Vapnik (2019). This new approach, referred to as intelligent-driven learning, relies on human understanding of the task at hand; by encoding the existing domain knowledge into predicates, it helps with model specification implicitly; hence, the model can achieve better accuracy using a small training set.

Inspired by this new paradigm, we further develop the theory of predicate-powered learning to propose the potential mechanism, and to justify the need for predicate-powered learning from two aspects: data complexity and model complexity. We suggest that the need for the large volume of data stems from data complexity. By providing predicates, we should be able to reduce the need for big data in most, if not all, learning tasks. Furthermore, we notice that the need for large models mainly comes from singletons in the data samples Brown et al. (2021), which are the sole representations of sub-distributions of data population; hence, memorization provided by large models become necessary. As we will show in later sections, this need can also be reduced, should a predicate-based scheme be adopted. Besides the theoretical justification of predicate-powered learning, we also conducted an experiment of learning invariants for an image classification task.

While the new paradigm of learning is still in development, there are models built implicitly utilizing the idea of predicates. Lake et al. (2015) shows that using a simple algorithm, Bayesian Program Learning, they can achieve human level performance for recognizing and generating handwritten complex alphabets. The key reason for this success, in our opinion, is that they effectively utilized human knowledge of how the alphabets are written, instead of solely relying on image data. We speculate that a similar method can also be adopted and solve the challenge proposed by Vapnik (2019), *i.e.,* getting state-of-the-art accuracy on MNIST dataset but only using 600 images in training. The ultimate question we are addressing is the following: can we find a middle ground between data and prior model specification?

KEY CONTRIBUTIONS

We further extend the theory of non-uniform learning with predicates, providing theoretical justifications of using predicates. From information theory perspectives, we show that using predicates could effectively reduce the need for large amounts of data. Furthermore, we show that predicates also have the potential to reduce the need for data memorization, and hence the model size. In summary, our key contributions are the following:

1. Proposal of a novel extended structure risk minimization rule with predicates.
2. First known effort to formalize the theory of predicate-powered learning.
3. Formally prove the data complexity reduction effects of predicates (reducing the need for big data).
4. Formally prove the model complexity reduction effects of predicates.

## 2 MAIN RESULTS

### 2.1 EXTENDED STRUCTURAL RISK MINIMIZATION

In classical learning theory, prior knowledge is encoded with a hypothesis function class. The need for predicates arises when we intend to define the hypothesis class based on certain properties. More specifically, the algorithm will utilize the predicates to differentiate the hypothesis classes. In this context, we assume that our initial hypothesis class is the union of countable sub-hypothesis classes. Effective predicates would then have the ability to differentiate those hypothesis classes.

In non-uniform learning, it can be assumed that each hypothesis class is uniformly learnable, and is assigned with a weight $w_i$ to reflect our prior belief (Shalev-Shwartz & Ben-David, 2009). We extend this idea to propose a structure risk minimization rule that we allow the weights to be adjusted via predicates (more on weights adjusting in next section). We adapt the following derivation from Shalev-Shwartz & Ben-David (2009).

Consider a hypothesis $\mathcal{H}$ that can be written as $\mathcal{H} = \bigcup_{i \in \mathbb{N}} \mathcal{H}_i$, where each $\mathcal{H}_i$ is uniformly learnable. We also assign a weight $w_i$ to each $\mathcal{H}_i$, *s.t.* $\sum_i w_i \leq 1$. By uniformly learnable, we mean that for each $\mathcal{H}_i$, there is an $m_{\mathcal{H}_i}(\xi, \delta)$, *s.t.* for any $\delta \in (0, 1), h \in \mathcal{H}_i$, we have

$$|L_D(h) - L_\mathcal{T}(h)| \leq \xi_i(m, w_i \delta)$$

where $L_D$ and $L_\mathcal{T}$ are the loss function based on data population $D$, and data sample $\mathcal{T}$, respectively; $\xi_i$ is the error rate function corresponding to $\mathcal{H}_i$:

$$\xi_i(m, \delta) = \min\{\xi \in (0, 1); m_{\mathcal{H}_i}(\xi, \delta) \leq m\}.$$

The function $m_\mathcal{H}(\xi, \delta)$ provides the minimal number of samples required to achieve an error rate less than $\xi$ with probability $1 - \delta$. This function also indicates a tradeoff between the error rate $\xi$ and the confidence $1 - \delta$: smaller the error rate, lower the confidence. The key idea of extended structural risk minimization is to trade $\delta$ with error rate $\xi$ via adjusting weights.

Since we don't require a disjoint union for $\mathcal{H}$, we can choose $\mathcal{H}_i$ that contains optimal $h$ with the smallest error. That is, for any $h \in \mathcal{H}$, we have

$$|L_D(h) - L_\mathcal{T}(h)| \leq \min_{i : h \in \mathcal{H}_i} \xi_i(m, w_i \delta).$$

It's clear that if $w_i$ is assumed to be increased, the error bound should decrease according to the error rate and confident tradeoff we mentioned above. We summarize this learning scheme as Algorithm 1.

Similar to SRM (Shalev-Shwartz & Ben-David, 2009), the ESRM can be used for non-uniform learning of every class. Particularly, we show that we can expedite the learning rate by providing useful predicates:

**Theorem 1** (Learning Rate of ESRM). *Given a non-uniform learnable hypothesis $\mathcal{H} = \bigcup_i \mathcal{H}_i$, assume there exists an unknown optimal model $f^* \in \mathcal{H}_n$, and sample data $\mathcal{T} \sim \mathcal{D}$. An effective*

---

**Pseudocode 1** Extended Structure Risk Minimization

---
1: **procedure** ESRM
2:     **Priors:** $\mathcal{H} = \bigcup \mathcal{H}_i$, *s.t.* $\mathcal{H}_i$ is uniformly learnable with $m_{\mathcal{H}_i}$, and initial $w_i$ *s.t.* $\sum_i w_i \leq 1$
3:     **Define:** $\xi = \{\xi \in (0,1) : m_{\mathcal{H}_i}(\xi, \delta) < m\}$ is the error controlled by the data size $m$ and $\delta$.
4:     **Input:** Data $\mathcal{T} \sim D$, predicates $\{\phi_i\}$, and confidence $1 - \delta$
5:     **Update:** $w_i \mathrel{+}= \Delta w(\phi_i, \mathcal{T})$
6:     **Output:** $h = \arg\min_{h \in \mathcal{H}} \left( L_{\mathcal{T}}(h) + \min_i \xi_{i:h \in \mathcal{H}_i}(m, w_i \cdot \delta_i) \right)$
7: **end procedure**

---

predicate $\phi$, *s.t. weight updating function* $\Delta w_n(\phi, \mathcal{T}) > 0$. *Let* $\hat{h} = SRM(\mathcal{T})$. *If* $\hat{h} \in \mathcal{H}_n$, *then* $ESRM(\mathcal{T}) = \hat{h}$. *Furthermore, for the same total error, the number of samples* $m$ *required for the learning reduced:* $m_{ESRM} \leq m_{SRM}$.

*Proof of Theorem 1.* In the structural risk minimization (SRM), we use pre-determined weights $w_i$ for $\mathcal{H}_i$, and the algorithm returns based on

$$\hat{h}_{SRM} = \arg\min_{h \in \mathcal{H}}[L_{\mathcal{T}}(h) + \min_j \xi_{j:h \in \mathcal{H}_j}(m, w_j \delta)].$$

where $\xi_{j:h \in \mathcal{H}_j}$ is the error rate function for $\mathcal{H}_j$ that contains $h$. In extended structural risk minimization (ESRM), by increasing the $w_i$ corresponding to $\mathcal{H}_i$ that contains optimal $h^*$, we increase $w_i \delta$, where $1 - \delta$ is the given confidence. Based on the confidence-error tradeoff, increasing $w_i$ causes $1 - \delta$ decrease, and error rate $\xi_i$ decrease as well (keep sample size $m$ fixed). Since the error rate function doesn't depend on any specific $h$, only $\mathcal{H}_i$. Then, the entire $\mathcal{H}_i$ has advantage in terms of minimization.

Hence, if the algorithm used in SRM chose $\mathcal{H}_i$ without weight update, the same algorithm under ESRM will still choose the same $\mathcal{H}_i$. Since the error rate function $\xi_i$ is not depending on $h$. The algorithm will still return the same $h$.

Due to confidence-error tradeoff, and we relaxed the confidence for $\mathcal{H}_i$ during weight update, we reduced $\xi_i$. Since the $h$ is the same, $L_{\mathcal{T}}(h)$ will stay the same; hence, the total error $L_D$ (generalization error; error in population) will reduce. Furthermore, if we keep the same error rate in SRM, we can increase the error function $\xi_i$ by decreasing $m$. Hence, for the same total error rate, $m_{ESRM} \leq m_{SRM}$. Equality may hold in a situation where $\xi_i$ is not strictly decreasing in $m$.     $\square$

## 2.2 PREDICATES FORMULATION

In this section, we will formulate the idea of predicates and its potential interaction with models. Based on ESRM rule proposed earlier, the crux of the predicate-powered learning is the hypothesis weights updating function $\Delta w(\mathcal{T}, \phi)$. This should reflect the likelihood of a hypothesis class under consideration. In the rest of this section, we will further develop this idea from the predicate-powered learning perspective.

The key idea in predicate-powered learning is to utilize predicates as a way to examine which of hypothesis classes are more plausible using existing knowledge. This would resemble probability density function estimation: we could use certain properties of the probability density function, such as mean and variance, as the testing criteria for the density function. In this case, mean and variance are served as predicates.

To make this idea more concrete, we can consider the pair of a model function and the predicate, and denote it as $\langle f, \phi \rangle$. Each pair should capture certain properties we care about and also have some knowledge about the function. **Note that, the pair $\langle f, \phi \rangle$ only denotes abstract interaction between the model and the predicates, such interaction should not be limited to the inner product.** Nonetheless, we will use the inner product as a special case in the subsequence sections for the analysis.

In the context of learning, as we are iteratively updating the model (*e.g.,* via gradient descent), the learning process can be considered as a sequence of $f_n$ such that

$$\langle f_n, \phi \rangle \to \langle y, \phi \rangle, \text{ as } n \to \infty,$$

where $y$ is the target labeling function. That's it, the properties of the model function gradually approaches the target labeling function. This formulation is consistent with Vapnik (2019), if we consider inner product in Hilbert space: rewrite this equation in discrete case, we have

$$\lim_{n \to \infty} \sum_i f_n(x_i)\phi(x_i) = \sum_i y_i\phi(x_i), \tag{1}$$

which coincide as the invariants admissive condition in Vapnik (2019). It's likely that we would enforce soft admissive conditions, and update the hypothesis weights accordingly as.

$$\Delta w_i \propto |\sum_i f(x_i)\phi(x_i) - y_i\phi(x_i)|,$$

where the index $i$ refers to the hypothesis class $\mathcal{H}_i$. However, it's not necessary to constraint the admissive conditions to the entire model. To generalize this idea, we consider model decomposition as $f = \sum_i f_i$. Then, for the learning task to be feasible, we would simply require each component of the model function to reflect class defining feature properties. To this end, using the linearity, we have

$$\langle \sum_i f_i, \phi \rangle = \sum_i \langle f_i, \phi \rangle \approx \langle y, \phi \rangle = \langle y, \phi_n \rangle + \langle y, \phi_m \rangle,$$

where $\phi = \phi_m + \phi_n$. Then, we simply set the admissive conditions to be

$$\Delta w_i \propto |\langle f_i, \phi_n \rangle - \langle y, \phi_m \rangle|.$$

It's worth noting that we don't require $\phi_m, \phi_n$ to be the same (although they should be related); we simply require the function properties align with the labeling function. With careful designs, we should be able to control the modeling behavior more granular. We can consider this as an implicit approach for model specification. In the classical wisdom of machine learning, we would choose a simpler model class if data is scarce. Nevertheless, the predicate presents another way of specifying a model besides prior model specification and data. By allowing the model evolve under the guidance of predicates, we could end up with fewer data and more sophisticated models.

## 2.3 Invariants in Hilbert Space

In this section, we will establish the connection between the admissive condition proposed in the last section with the concept of invariants. *Both of them are special case of controlling the hypothesis space of learning,* i.e., *different interaction between $f$ and $\phi$.* To facilitate the analysis, we only consider the Reproducing Kernel Hilbert Space (RKHS). While we believe that the analysis can be extended to more generalized settings, we would leave that to future work. Let a generic perturbation of input data $x \in X$ be $\phi$, the model $f$ is said to be invariant against the perturbation $\phi$, should the following condition holds:

$$f(\phi(x)) \approx f(x), \tag{2}$$

where $f : X \mapsto \mathbb{R}$, we don't force strict equality to allow practically negligible difference. Then, we can consider the LHS of eq. (2) as a linear functional, induced by $f$, acting on the perturbation $\phi$. That is

$$f(\phi(x)) = L_f(\phi; x) = \langle \phi, \tilde{f} \rangle = \langle \tilde{f}, \phi \rangle,$$

where second equality is by Riesz representation theorem, last equality holds since we are working on real space. Next, for the RHS of eq. (2), we have

$$f(x) = \langle f, K(\cdot, x) \rangle = \langle f, \tilde{\phi} \rangle \approx \langle y, \tilde{\phi} \rangle,$$

where $y$ is the target labeling function, $K$ is the reproducing kernel, and we rewrite $K(\cdot, x)$ as $\tilde{\phi}$. Hence, by imposing

$$\langle \tilde{f}, \phi \rangle \approx \langle y, \tilde{\phi} \rangle,$$

we implicitly learn a function that is perturbation invariant, but $\tilde{f}$ learned using statistical invariants is not the perturbation invariant $f$ at the beginning of the analysis. It's worth noting that we only consider more general cases in the last section where we don't impose that $\phi$ and $\tilde{\phi}$ to be the same. Lastly, the analysis in this section only means to draw connection between the concept of Statistical invariants introduced in Vapnik & Izmailov (2019) (as well as the admissive conditions mentioned earlier) and more common concepts of invariants in the literature, such as Bloem-Reddy & Teh (2020). These two concepts should not be considered as equivalent but related.

### 2.4 CONNECTION TO INFORMATION THEORY

We argue that providing predicates this way can help with establishing the equivalent classes. Since the predicates will shrink the hypothesis class, there will eventually be data points that cannot be separated by the admissible hypothesis class. Hence, these unseparable data points form their respective equivalent classes.

We will show that using predicates is beneficial in reducing the perceived data complexity and required model complexity. We postpone the detailed discussion and terminology definitions to Section 4 and Section 5. We consider the data complexity as the inherent difficulty of classification. The data complexity is related to the minimal number of data samples we need to achieve the goal of learning. The key idea is that if a predicate can describe the variation or the characteristics within an equivalent class, we would not need more than one example of this class. Hence, we could use the intelligence provided by the predicates to reduce the need for data.

Similarly, we can also consider the model complexity as the minimal complexity required to solve a task. This would naturally be related to the number of equivalent classes in the data. Using predicates can then reduce the model complexity required. The concept is closely related to data complexity. Additionally, using predicates in ESRM can also directly favor simpler hypothesis function classes.

## 3 RELATED WORK

### 3.1 *Learning using Statistical Invariants*

Our work is essentially inspired by Vapnik & Izmailov (2019); Vapnik (2019) on the new paradigm of learning, dubbed learning using statistical invariants or complete learning theory. The key idea is using predicates (statistical invariants) to facilitate the learning process. Doing so will utilize both strong and weak convergence in the Hilbert space; hence, it's complete. The original method largely relies on constrained optimization. This, however, is less practical in deep learning, and has not received wide adoption. Our work extends the idea of *Learning Using Statistical Invariants*, provides further theoretical justification for the need of predicate-powered learning.

### 3.2 CONCEPT LEARNING

Predicate-powered learning is also related to concept learning. Instead of only predicting the final targets or labels, concept learning tasks also try to learn the concepts presented in the input data. Lake et al. (2015) proposed a Bayesian based algorithm to learn handwritten complex alphabets, it shows human level performance in recognizing these alphabets. While the proposed model does appear to be limited to the task of handwritten alphabets, we consider it to be an excellent example of utilizing human knowledge of the alphabets to reduce the need for big data and large models. While the theory provided in this work is not directly for concept learning, we can still consider the

learned concepts as the invariants. Therefore, our experiments in this paper can also be considered as concept learning.

Additionally, Mordatch (2018) proposed using energy-based models to learn scene concepts, such as background, simple geometry objects, etc. Koh et al. (2020) proposed a bottleneck approach to learning intermediate concepts along with the final targets. However, these methods, mostly data-driven, did not take advantage of existing knowledge.

### 3.3 INVARIANT NETWORKS

There have been many efforts to utilize invariants by designing specialized neural network architectures. The Convolutional Neural Network (CNN) (LeCun et al., 1995) is the first known neural network to take advantage of spatial invariants' information on image data. Group Equivariant Convolutional Network (Cohen & Welling, 2016) further extends the network architecture to exploit the symmetries, such as rotation invariant. Among many other further developments on equivariant network, Satorras et al. (2021) extends the idea to graph neural networks, Romero et al. (2020) adds attention mechanism to equivariant network to learn meaningful relationships among features. Bloem-Reddy & Teh (2020) further characterizes the structure of invariant or equivariant networks.

### 3.4 OTHERS

Closely to predicates, physics-informed deep learning (Raissi et al., 2017) has been combining differential equations with neural networks as a successful way to incorporate domain knowledge, while predicate-powered learning is designed for applications in which differential equations are not applicable. Learning invariants has also been seen in other domains, such as reinforce learning (Zhang et al., 2020), to reduce the computation cost of reconstruction. Furthermore, this work also takes inspiration from Jin et al. (2020), which quantifies generalization error of learning tasks. Additionally, the memorization problem of neural networks has been studied by Brown et al. (2021); Arpit et al. (2017); Makarychev et al. (2020); they raise the question of what role memorization plays in the learning process, Brown et al. (2021) further suggest memorization in some situations might be necessary for near-optimal solutions.

## 4 DATA COMPLEXITY

In this section, we will investigate the effects of predicates in data domain, specifically, in terms of data complexity. As the goal of a learning is to construct a mapping from data domain to label domain, the higher the data complexity, the higher the difficulty of the learning task (see Appendix A.1 more detailed discussion).

In our theory, we assume the predicates can provide information that is relevant to the targets but hard or infeasible to acquire via data alone, and also difficult to express in prior model specification. To make the theoretical derivation more approachable, without much loss of generality, **we assume the data population $D$ constitutes a *separable* $d$-dimensional hypercube in $\mathbb{R}^d$.** Additionally, we assume information is dense in the sense that it is concentrated at an $\epsilon$-ball for data point $x_i$. This means that the information in the population can be recovered from countable many data samples $\{x_i\}$, and without further conditions, we will assume each $x_i$ possess different information. Later, we will show that using predicates, we can build connections among those data point $x_i$.

The learning task is to recover the information related to the *labels* in population $D$, using a finite sample $\mathcal{T} = \{x_i\}$. In our analysis, we will use the concept of cover to describe retrieving the information from the data population $D$.

**Definition 1** (Cover). *Let $D$ be the data population, and $\mathcal{T}$ be a sample,* i.e., *set $\{x_1, x_2, ..., x_n\}$. Then, we consider $\mathcal{T}$ forms a cover $\mathcal{B} = \bigcup_{x_i \in \mathcal{T}} \left( B(x_i, \epsilon) \bigcap D \right)$ for $D$. where $y_k$ is the labeling function for the task.*

In the cover defined above, the goal is not to recover all the information from $D$ or the subset of $D$, but the information related to the labeling $y_k$ conditioning on $D$. Naturally, if we have the data point $x_i$ in our sample, we should have the information in $x_i$, including the information related to

$y_k$. Later we will see, if we can establish the equivalent relations, we can even know the information regarding $y_k$ beyond the data sample $\mathcal{T}$. This is because the extra information in $x_i$ has little to do with the labeling $y_k$. For instance, if one is aware that insects possess six legs and three body parts, they can readily discern that spiders or centipedes are not insects without having previously acquired knowledge about them. *This mechanism resembles sufficient statistics, if we know sufficient statistics, we can abandon the data samples without losing the information we need.* the context of predicates, which will be crucial in further analysis. Then, it's clear that all we need is some guidance to discern which part of the data is relevant to the task.

We first list some key definitions related to our analysis.

**Definition 2** (Labeling function). *For input $x_i \in D$, we have a labeling (predicting) function $y$ that returns Bayes optimal prediction. We abbreviate $y_i := y(x_i)$.*

**Definition 3** (Conditional Probability). *Consider $e_i \subset D$ and we denote the probability of predication as*

$$P(y(x_i) = \gamma | e_i) = \frac{P(y^{-1}(\gamma) \cup e_i)}{P(e_i)}$$

*where $y^{-1}(\gamma)$ is the pre-image of the labeling function. We abbreviate this notation as $P(y_i | e_i)$, since we do not care for any specific $\gamma$, we only care for the relations between different probabilities.*

**Definition 4** (Predicates). *We define a predicate as a mapping $\phi : D \mapsto \mathbb{R}^n$ for some $n \in \mathbb{N}$. Then, there is an operator $\wp$, s.t. $\wp(y, \phi) = \tilde{y}$. we denote*

$$P(y_i; \phi) := P(y(x_i) = \gamma; \phi) := P(\tilde{y}(x_i) = \gamma)$$

*for some $\gamma$. The operator will always exist, since we can always choose the trivial operator that simply returns $y$.*

We assume $y$ is the optimal predictor for the given information, and an effective predicate can improve the $y$ by utilize information not available in the data sample $\mathcal{T}$, *i.e.,* domain knowledge. We call two subsets of the data equivalent, if they provide some amount of information for our learning task, *i.e.,* $y$ will make the same predications with the same likelihood. We formulate this equivalent notion as below.

**Definition 5** (Equivalent Relation). *Let $D$ be the data population, $\mathcal{T} \subset D$ be the data sample, and $e_i$ be an $\epsilon$-ball around $x_i \in D$. For a labeling r.v. $y_k$, then, we say $e_i$ and $e_j$ are equivalent, denoted as $e_i \sim e_j$, if and only if every prediction conditioning on $e_i, e_j$ is the same, up to some negligible margin:*

$$\forall \xi > 0, \ \sup_k |P(y_k | e_i) - P(y_k | e_j)| < \xi$$

*Then, we say a predicate $\phi$ is effective, if it helps to establish equivalent relations in the sense that*

$$\forall \xi > 0, \ \sup_k |P(y_k | e_i; \phi) - P(y_k | e_j; \phi)| < \xi \tag{3}$$

*where $e_i \subset \mathcal{T}$ and $e_j \subset D$ for some sample $\mathcal{T}$ drawn from a data population $\mathcal{D}$. Additionally, when $e_j \subset D \backslash \mathcal{T}$, we call it **type-I** predicate, and when $e_j \subset \mathcal{T}$, we call it **type-II** predicate.*

Based on the formulation above, we will see that type-I predicates will help reduce the data samples requirement, and type-II predicates will help reduce the model complexity. Note that these two types of predicates are not mutually exclusive; *it would be desirable that the predicates are of both types.* Next, we will construct a Pseudo-measure in the context of equivalent relations in the sense of Definition 5. This measure will help to quantify the information related to the task recovered by the cover (or data samples).

**Definition 6** (Pseudo-Measure of Equivalent Classes). *We define $\mu$ as a pseudo-measure of $\omega \in \mathfrak{S}(D)$, where $\omega = \bigcup_i e_i$, if and only if*

$$\mu(\omega) = \mu_L(\bigcup_j e_j : e_j \sim e_i, e_i \subset \omega, e_j \in \mathfrak{S}(D)),$$

*where $\mu_L$ denotes the Lebesgue measure in $\mathbb{R}^d$, $\sim$ denotes the equivalent relations defined in Definition 5, and $\mathfrak{S}$ denotes the sigma algebra.*

We notice that the construction of $\omega = \bigcup_j e_j$ is not unique: there are multiple ways to form such a union. However, since we essentially form countable partitions of the data population in terms of equivalency, hence, choosing different unions will not change the measure. Finally, we define cover ratio.

**Definition 7** (Cover Ratio). *Let $D$ be the data population, $\mathcal{T} \subset D$ be the sample, which forms a cover $\mathcal{B} = \bigcup_{x_i \in \mathcal{T}} \left( B(x_i, \epsilon) \bigcap D \right)$, then define the cover ratio as*

$$\rho = \frac{\mu(\mathcal{B})}{\mu(D)}.$$

The cover ratio indicates the amount of information that can be recovered from the data sample. Hence, it's directly related to the expected accuracy of the model. Thanks to the predicates, we can retrieve the task specific information more efficiently.

**Theorem 2** (Expected Accuracy). *Let $D$ be the data population, $\mathcal{T} \subset D$ be the sample, which forms a cover $\mathcal{B} = \bigcup_{x_i \in \mathcal{T}} \left( B(x_i, \epsilon) \bigcap D \right)$. Then, the expected accuracy $\pi$ of a well-trained model is equal to the cover ratio $\rho$. Furthermore, keeping everything else constant, using effective predicates will not reduce $\pi$, and if predicates are type-I effective, then they cause $\pi$ increase.*

*Proof.* **All proofs can be found in the Appendix B.** □

Consequently, we can maintain the same accuracy with a reduced amount of data.

## 5 MODEL COMPLEXITY

One crucial factor that has an impact on the model complexity is the presence of singletons within the data sample. As they are the sole example of the sub-distribution of the population, the model has to memorize these singletons; this phenomenon makes small models or model compression inevitably suboptimal (Brown et al., 2021).

This, however, can be resolved by establishing equivalent classes via predicates, which will result in a simpler mixture of sub-distribution structure in the data, and reduce the chance of the appearance of the singletons (up to the equivalent relations). Therefore, the memorization requirement will be relaxed without suffering from the loss of predictive accuracy. Next, we make this precise.

To precisely quantify the complexity of the model, we define the cover entropy of a learnable mapping. First, we define the concept of sub-cover in this context.

**Definition 8** (Sub-cover). *For any random sample $\mathcal{T}$ from data population $D$, and $\mathcal{T}$ constructs a cover $\mathcal{B}$ for $D$. $\tilde{\mathcal{B}}$ is a sub-cover of $\mathcal{B}$, if and only if*

$$\mu(\mathcal{B}) = \mu(\tilde{\mathcal{B}}),$$

*where $\tilde{\mathcal{B}}$ is constructed by a smaller sample $\tilde{\mathcal{T}}$, i.e.,*

$$|\tilde{\mathcal{B}}| \leq |\mathcal{B}|, \tilde{\mathcal{T}} \subset \mathcal{T},$$

*where $|\mathcal{B}|$ refers to the cardinality of $\mathcal{B}$, which is the size of $\mathcal{T}$ that forms $\mathcal{B}$. We also consider cover $\mathcal{B}$ as a sub-cover of $\mathcal{B}$ itself.*

**Definition 9** (Cover Entropy). *Consider a sample $\mathcal{T}$ from a data population $D$, and a data cover $\mathcal{B}$ formed by $\mathcal{T}$. Then, there is a sub-cover of $\mathcal{B}$. Let $\mathcal{B}_0$ be the smallest sub-cover, s.t.*

$$|\mathcal{B}_0| = \inf_{\tilde{\mathcal{B}} \subset \mathcal{B}} |\tilde{\mathcal{B}}|,$$

*where $|\mathcal{B}|$ denotes the cardinality of $\mathcal{B}$. Then, the cover entropy of a learning task on the data sample is defined as*

$$H^c(\mathcal{T}) = \log(|\mathcal{B}_0|).$$

We define the model complexity as the optimal model complexity: they are the simplest models required to utilize the information fully in the data sample for the learning task. Any simpler model will not be optimal in terms of expected accuracy.

Before we introduce the main theorem for this section, we show that if predicates are effective, we reduce the chance of singletons; hence, this will alleviate the need for large models.

**Proposition 1** (Singletons (up to equivalent relations)). *Consider a sample $\mathcal{T} \in \mathfrak{S}(D)$, let $N^s$ be the number of singletons in the data sample $\mathcal{T}$, then*

$$\mathbb{E}N_\phi^s(\mathcal{T}) < \mathbb{E}N^s(\mathcal{T}),$$

*where $\phi$ is an effective predicate.*

Finally, we show that if we introduce effective predicates in our learning algorithms, we can reduce the cover entropy, *i.e.,* the lower bound for the model complexity. This allows a simpler model to be learned.

**Theorem 3** (Complexity Reduction). *Consider a sample $\mathcal{T}$ from a data population $D$. If there is an effective type-II predicate $\phi$, then, we have*

$$H_\phi^c(\mathcal{T}) < H^c(\mathcal{T}),$$

*where $H_\phi^c$ is the cover entropy after applying the predicates, while maintaining the same expected predictive accuracy $\pi$.*

This theorem and Theorem 2 depicts different mechanisms of the predicates: 1) finding redundant information within the sample, or 2) finding wider relationship with the data population.

## 6 CONCLUSION

In this paper, we explore the idea of predicate-powered learning as an emerging paradigm of learning and propose an extended version of structure risk minimization. We consider it has the potential of breaking the current trend of "brute force" learning method. Specifically, we carefully investigate the potential role of predicates in the learning process, and we propose two primary justifications for the necessity of predicates, which stem from two essential aspects of the learning process, namely, data and model. From a general information theory perspective, we show that predicates have the potential to reduce the amount of data and the size of the model for a given task using invariants introduced by existing knowledge.

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

# A  DATA & MODEL COMPLEXITY

## A.1  REDUCING INFORMATION USING INVARIANTS

At its core, we posit that the primary objective of incorporating invariants is to reduce the information complexity. Moreover, we surmise that there are multiple ways that the predicates can interact with the model building process. For any given learning task $\mathcal{L}$, there are three major components: the data sample $\mathcal{T}$, the learnable mapping $f$, and the learning algorithm $\mathcal{A}$. The goal of such a task would be to optimize $f$, *s.t.*

$$\hat{f} = \mathcal{A}(f, \mathcal{T}) \approx f^*,$$

where $f^*$ is theoretical best model, or true data generating function. Hence, we have three potential mechanisms that predicates can be involved, *i.e.,* the predicates can interact with

a) **data sample $\mathcal{T}$.**  transform the input data sample, such that the irrelevant information is no longer present in the data.

b) **learnable mapping $f$.**  construct the mapping utilizing the predicates, such that the mapping will ignore the irrelevant information. Convolution networks can be considered as an example of this kind.

*c)* **algorithm** $\mathcal{A}$. tune the mapping under the guidance of the predicates. Using constraints in the loss function as in Vapnik (2019) can be considered as an example of this kind.

We can see these three mechanisms are not independent of each other; rather, they are theoretically equivalent. However, in practice, it's not hard to imagine they would be task-dependent: one mechanism would likely work better than the rest.

In the case of predicates interacting with data, we can consider transforming the input data $X$ into a quotient set using predicates $\phi$, *i.e.*, $X_{|\phi}$, then we have $\mathfrak{S}(X_{|\phi}) \subset \mathfrak{S}(X)$. To this end, we can consider the information contained in the targets as $\mathfrak{S}(X_{|f})$, where $f$ is the true model.

Then, we can interpret the learning task as finding the ultimate equivalent classes that all data with the same label or target are equivalent. This leads to our definition for effective predicates.

From an information theory perspective, the modeling function is to reduce the information represented by the data to the information represented by the targets. For complicated data, this can be challenging. If the predicates can significantly reduce the information, as a part of the model, the learning process would be easier.

One example of predicates can be the sufficient statistics: it helps the model by providing key insights into the data, while getting rid of redundant information. However, we notice this can be overly strong in practice in most non-trivial problems, where we're unlikely to obtain such information.

## B PROOFS

### B.1 DATA COMPLEXITY

Theorem 2

*Proof.* Since a well-trained model will utilize available information, and cover ratio represents the ratio of information recovered from the data population. Hence, for any data sample drawn from the population, we can expect accuracy to be $\pi = \rho$.

By definition of $\mu$ in Definition 6, $\sup_{\omega \in \mathfrak{S}(D)} \mu(\omega) = \mu(D)$, therefore, $\mu(D)$ will not decrease.

Applying type-I predicates will introduce new equivalent relations; therefore, there is $e_j \subset D \backslash \mathcal{T}$ with positive measure, *s.t.*

$$e_i \sim_\phi e_j, e_i \not\sim e_j$$

where $e.$ are $\epsilon$-balls around data points, $e_i \subset \mathcal{B}$, and $\sim_\phi$ is the equivalent relation founded by the predicate. Then,

$$\mu_\phi(\mathcal{B}) = \mu(\mathcal{B}) + \mu_L(\bigcup e_j : e_j \sim_\phi e_i, e_j \in \mathfrak{S}(D \backslash \mathcal{B})) > \mu(\mathcal{B})$$

where $e_i \subset \mathcal{B}$, $\mu_\phi$ denotes the pseudo-measure after the predicate, $\mu_L$ denotes the Lebesgue measure in $\mathbb{R}^d$, $\mathfrak{S}$ denotes the sigma algebra. Hence, if predicates are effective, $\rho = \mu(\mathcal{B})/\mu(D)$ will increase, so does $\pi$. $\square$

### B.2 MODEL COMPLEXITY

Proposition 1

*Proof.* Since the sample $\mathcal{T}$ is formed independent of the predicates, we can switch the order of them. First, before we have any predicates, data samples may still naturally form some partitions of equivalent classes: they are the sub-distributions. Then, let's consider the worst case when all data points are singletons. This configuration has non-zero probability as long as the size of $\mathcal{T}$ is smaller than the number of sub-distributions.

Next, if we apply predicates of any kind to establish new equivalent relations. We will reduce the number of partitions/sub-distributions. Therefore, the expected number of singletons will naturally be reduced. $\square$

Theorem 3

*Proof.* According to Theorem 2, we only need to keep the cover ratio $\rho$ in order to keep the expected accuracy $\pi$.

By definition, $\mathcal{B}$ is the union of the $\epsilon$-ball $e_i$ around the data point $x_i \in \mathcal{T}$. If the predicate is type-II effective by Definition 5, then, at least one new equivalent relation is found, *s.t.*

$$e_j \sim_\phi e_i, e_j \nsim e_i$$

where $\sim_\phi$ denotes the relation found by the predicate $\phi$, and $e_j \subset \mathcal{T}$. Because of this, we can construct a sub-cover $\tilde{\mathcal{B}}$ *s.t.* $e_j \not\subset \tilde{\mathcal{B}}$ and $\mu(\tilde{\mathcal{B}}) = \mu(\mathcal{B})$. This means, $|\tilde{\mathcal{B}}| < |\mathcal{B}|$, hence,

$$H_\phi^c(\mathcal{T}) < H^c(\mathcal{T}).$$

$\square$

