# OpenReview forum: "Towards Predicate-powered Learning"
_ICLR.cc/2024/Conference — ICLR 2024 Conference Withdrawn Submission_

### Official Review · Reviewer_afMq · 2023-11-01

**Soundness:** 3 good
**Presentation:** 3 good
**Contribution:** 2 fair
**Rating:** 5
**Confidence:** 2

**Summary:**

In this paper, the authors present an extension to the structure risk minimization.
Here the authors present proofs for the impact of effective predicates on learning for data and model complexity.
The authors then provide an empirical evaluation for direct and statistical invariants.

**Strengths:**

Starting from effective predicates, the authors then build proofs showing a reduction in data and model complexity.

**Weaknesses:**

From the paper, it does not seem like finding effective predicates is easy, nor integrating them into the learning algorithm.

The empirical examination does not properly support the claims of the paper, nor serve as a good example of the proofs working and this is the weakest part of the paper.

**Questions:**

Does it make sense and invert the problem, i.e., ask, what data points are needed for learning based on the predicates? This could be useful for use cases where exploration is feasible.

---

> ### Author Response · Authors · 2023-11-21
>
> Thank you for your feedback. We apologize for the editorial quality issues. We carefully revised the
> paper, most concerns, if not all, should be addressed.
> Regarding your questions:
> 1. We didn’t consider that direction, since in most learning tasks data is given, and out of our
> control. However, we assume such direction is feasible in active learning.

---

> ### Comment · Reviewer_afMq · 2023-11-22
>
> After reading the other reviews, especially Reviewer oViS. I decided that I will keep my score.

---

### Official Review · Reviewer_ufoZ · 2023-11-03

**Soundness:** 2 fair
**Presentation:** 2 fair
**Contribution:** 2 fair
**Rating:** 3
**Confidence:** 3

**Summary:**

This paper may have a good idea, but is not ready for publication. There are too many assumptions made by the authors that are not made explicit. (Caveat: I am not a researcher in the area of the paper, but I should be able to understand it). The experiments are not up to the standard of a top conference. They seem to show that the proposed idea does not work.

**Strengths:**

see above

**Weaknesses:**

First, I think what you call predicates should be called meta-predicates, as they are predicates about the data, not predicates in the data (e.g., as you might find in relational data). I had assumed that you meant predicate as in logic that is a Boolean function, rather than as in the predicate of a sentence. In any case you need to tell us what a predicate is.

There are lots of problems with notation:
In pseudocode 1 (and surrounding text). Are the w_i are probabilities? Are the hypotheses disjoint? But then in theorem 1, the weight update is assume to be positive; surely it can't be positive for all hypotheses. (I think the problem is that there is an explicit quantification that is not given. For it mean "for all n" or "there exists n"? - as I don't see what n is). In the equation after equation (1), \Delta w_i is always positive; w_i needs to be explained better,

You are assuming much more of the reader than can be assumed. You tell us that <f,\phi> is a pair. And then a very strange sentence in bold; I'm not sure why we would assume any interaction between f and \phi; it is just a pair.  I think you mean a function of f and \phi with some properties that you don't state. It is then fine to use the inner product as example.

In equation (2) why is it only "approximately equal"? Surely it should be =

Please don't say "which resembles human learning" without saying how it resembles human learning and providing evidence (eg, a reference) that says that human learning is like that.

On the bottom of page 5, how can there not be "countable many data points"? (I'm having trouble with "without much loss of generality; I'm thinking of one of the dimensions is time, and we want to predict the future from that past; I'm guessing that is a case that is not covered ;^)

On the top of page 6, "Naturally, if we have..." seems to imply there is no noise in the data. If that is an assumption it should be made explicit.

On definition 2, please don't use epsilon for two different concepts -- I see now that the epsilon in expsilon-ball is a different font than the < epsilon  (they have to be different as they different units), but it was very confusing. This should be called an equivalent relation as it is not transitive.

Definition 3 doesn't make sense. I don't think you want "Let".

Definition 5 doesn't make sense. What is "they"? I can't see a definition of sub-cover here. Definition 6 seems to also make a claim "Then, there is a..." which isn't obvious.

Why is the union not unique? I don't think I understand what it is then.

Page 8, at the bottom "same prediction" seems to imply that y is discrete. If that is the case, it should be made explicit. (Or what does "same" mean; do you want some epsilon? )

The experiments are not up to the standards of a top conference. The experiments are the average of 10 or 5 runs. No error bars are given, but I would expect the variance to be so high that the results are meaningless. (All the numbers in table 1 look the same to me).

**Questions:**

There is no explicit definition of a predicate. I think it is a function on examples. But isn't that a property? What makes these properties special in that they can be treated differently from the other properties in the data?

---

> ### Author Response · Authors · 2023-11-21
>
> Thank you for your feedback. We apologize for the editorial quality issues. We carefully revised the
> paper, most concerns, if not all, should be addressed.
> Regarding some of your feedback,
> 1. The term predicate is first used in Vapnik’s paper. We use the same terminology to be consistent
> with existing literature.
> 2. We updated the paper to include a predicate definition.
> 3. w_i is the weights for each hypothesis class H_i. Since our predicate indicating which
> hypothesis should be favored, the weight update should be positive. We can renormalize the
> weight after update.
> 4. < f, phi> denote the interaction of them. Inner product is an example of such interaction.
> 5. We are not sure your concern with “countable many data points”. We simply assume the
> information can be recovered with countable many data points.
> 6. We didn’t assume no noise in the data, our analysis is agnostic regarding noise. In the paper, we
> simply state if we have x_i in our sample, we have the information in that data point, whether it
> contains noise or not.
> 7. Our equivalent relation is transitive. Our definition of equivalent relation resembles the
> equivalent class concept in quotient space.
> 8. Sorry for the wording issue in some definitions, e.g. “there is…”. We didn’t intend to claim any
> results by them, but pointing to the concept. We updated the paper to clear this confusion.
> 9. We meant to state that the construction of the union is not unique, not the union itself.
> 10. Although transferable, we focus on classification task in our analysis.
>
> Regarding your questions:
> 1. We have updated the paper with a definition for predicate. The predicate points out the invariants
> in the data via domain knowledge. Invariance property are important in the sense they tell the
> model where to look.

---

### Official Review · Reviewer_kdak · 2023-11-05

**Soundness:** 2 fair
**Presentation:** 1 poor
**Contribution:** 2 fair
**Rating:** 3
**Confidence:** 4

**Summary:**

This paper develops a theory to prove that predicate-powered learning reduces data complexity to build models (not many instances are needed for training) and reduces model complexity.

**Strengths:**

- Attempt to formalize the use of predicates on learning tasks
- Attempt to prove data and model complexity reduction

**Weaknesses:**

- Lack of background on some of the concepts used
- Domain knowledge is hard to code and your work depend on domain knowledge
- Only one experiment with one dataset.

**Questions:**

I found this paper very hard to read maybe because the concept of predicate used in this paper does not match the concept of logic predicate (knowledge) I have in mind. The title of this paper mentions the use of predicates. I could not find any predicate example and how they are used/encoded in this work. It'd be ideal to give an example. Authors mention mean and variance, but these can be well used for numerical data. What about categorical data, where predicates are actually useful?

Theorem 1: talks about experditing the learning rate by providing useful predicates, however, the proof mentions number of samples required for learning of SRM being smaller than for ESRM. What is the relation between the reduction in number of samples and learning rate (smaller data?) and how about quality of the model with the reduction in data?

In Pseudocode 1, line 4, what are the predicates? How do you obtain them?

S2.2: ESRM rule proposed earlier: are you talking about Pseudocode 1? Is it a rule? Or are you talking about some rule used in the original SRM?

There are some works that embed expert knowledge to learn new models (e.g., https://www.ncbi.nlm.nih.gov/pmc/articles/PMC4525246/ or https://www.jmlr.org/papers/volume17/15-444/15-444.pdf). Knowledge can be embedded through propositionalization or using (first-order) relational models. I guess one of your contribs is to formalize the way knowledge is embedded. However, it looks like your method is limited to numerical or image data.

Typos etc:
to differentiate the among those hypothesis classes. --> to differentiate among those hypothesis classes.

a structure risk minimization rule that we allow the weights -->  a structure risk minimization rule that allows (will allow?) the weights

hat the model should predicate the same --> hat the model should predict (??) the same

---

> ### Author Response · Authors · 2023-11-21
>
> Thank you for your feedback. We apologize for the editorial quality issues. We carefully revised the
> paper, most concerns, if not all, should be addressed.
> Regarding some of your questions:
> 1. This paper meant to be a conceptual work, we refer to more concrete examples in related works.
> 2. Predicate are useful in the sense of guiding the algorithm which functions space to search for. So,
> predicates are useful in most if not all learning tasks, regardless of data types.
> 4. The learning rate is inversely proportional to the sample used in the learning process. The quality
> will be the same. see the updated version of theorem 1.
> 4. We updated the paper to define predicate explicitly. As to obtaining predicate is not within the
> scope of the paper. Predicates are assumed to be constructed using domain knowledge of the task.
> 5. ESRM rule simply refers to the ESRM learning approach.
> 6. We are sure why you would consider this is limited to numerical or image data. This supposed to
> be general purpose.

---

### Official Review · Reviewer_oViS · 2023-11-08

**Soundness:** 1 poor
**Presentation:** 2 fair
**Contribution:** 2 fair
**Rating:** 3
**Confidence:** 3

**Summary:**

The paper considers extending work of Vapnik on learning using statistical invariants. Particularly, they propose an extension of structural risk minimization in which the weights $w_i$ associated to classes $H_i$ are updated to favor those $H_i$ that are closer to satisfying desired invariants. They also try to formalize how predicates can divide the data into equivalence classes, leading to reduced data and model complexity requirements.

**Strengths:**

The idea of adjusting weights in SRM to reflect adherence to invariants offers an interesting and straightforward approach for integrating prior knowledge. And it seems worthwhile to explore quantifying how predicates can reduce the need for additional data.

**Weaknesses:**

Various typos (e.g. p.2 “the ability to differentiate the among those hypothesis classes”, p.3 “control the modeling behavior more granular”) and nonstandard terminology (e.g. "structure risk minimization", "equivalent relation") make the paper hard to read.


The weight update scheme for ESRM is not rigorously developed:

-The pseudocode for ESRM is hard to follow: Perhaps the intention was for $\epsilon_i$ to be the minimum of the set rather than the set itself. Since $h$ may belong to multiple $H_i$, the objective is also ambiguous as written.

-Theorem 1 seems to only handwave that if we are able to update SRM weights to favor the class $H_i$ containing an optimal model, then we expect the performance of SRM to improve. The subsequent section does not adequately clarify how this would be achieved.

Section 2.3 is hard to contextualize with the rest of the paper:

-The proposed aim is to connect statistical invariants as defined by Vapnik to the concept of invariants elsewhere (e.g. models that require the same output on the orbit of $x$ under some group action). How the analysis achieves this is fuzzy, perhaps due to a lack of definitions. Might be nice to look at the recent work https://proceedings.mlr.press/v128/vapnik20a.html, which has more examples of predicates describing symmetries.

As written, the definitions and proofs in section 4 and 5 are not rigorous or correct. For example, in Definition 1 the meaning of $P(y_k | \mathcal{B})$ is not clear, perhaps intended to be an expected value over $\mathcal{B}$. It is also confusing that the definition seems to be trivially satisfied by setting $\tilde{D}=\mathcal{B}$. This confusion is exacerbated in Definition 2, where the meaning of $P(y_k | e_i; \phi)$ is ambiguous. The proof of Theorem 2 is not rigorous, affecting the proof of Theorem 3. Similar problems with Proposition 1.

The experiments in section 6 would be improved by graphics comparing the baseline to the invariant augmented models, and employing some form of uncertainty quantification. The connection to ESRM could also be better developed.

**Questions:**

-In section 2.2, it is suggested to update weights for $H_i$ proportional to $|\sum f(x_i) \phi(x_i) - y_i \phi(x_i)|$. Is $f$ an arbitrary element of $H_i$? The purpose of subsequently decomposing $f$ and $\phi$ is also unclear.

-In section 2.3, could you clarify the definition and domain of the linear functional $L_f$? It reads as if $L_f$ might map $\phi$ to composition $f \circ \phi$, but it is not clear to me why this would be linear.

-In section 4, what precisely are $P(y_k |\mathcal{B})$ and $P(y_k|e_i; \phi )$? Why do you assume the difference is less than $\varepsilon$ for all $\varepsilon$, rather than equal to zero?

-In the proof of Proposition 1, why do you require the axiom of choice when we're dealing with finite samples?

---

> ### Author Response · Authors · 2023-11-21
>
> Thank you for your feedback. We apologize for the editorial quality issues. We carefully revised the
> paper, most concerns, if not all, should be addressed.
> Regarding some of your feedback,
> 1. We updated the more details of ESRM algorithm to make more clear and rigorous, including the
> theorem 1.
> 2. More definitions are introduced to make the original definitions and proposition clearer.
>
> Regarding your questions:
> 1. Yes, decomposing of f means to tackle the problem piecewise.
> 2. L_f is a mapping from phi to f(phi(x)). It’s linear in phi as it can be rewritten as inner product
> with phi by Riesz representation theorem.
> 3. We updated the paper to explain the these concepts in details. P(y_k|B) are probabilities of a
> certain prediction given B, likewise, P(y_k|e;phi) are probabilities of a certain prediction given e_i
> and predicate.
> 4. We remove the requirement for axiom of choice.

---

> > ### Comment · Reviewer_oViS · 2023-11-22
> >
> > I have read the authors updates but they inadequately address my questions and concerns (e.g. means of achieving ESRM weight update are still not clear, use of Riesz representation is incorrect, etc.)

---

### Author Response · Authors · 2023-11-21

Thank you for everyone's feedback. We carefully revised the paper, most concerns, if not all, should be addressed. We realized the issue regarding the experiments, as our theory focuses on the abstraction of the learning process, we removed the experiments and provide more details on our theory.